# Improving the prediction of potato yield gaps: Solanum-model parameterization and evaluation in southwestern China

Ying Wang[1,2], Muhammad Abdul Rehman Rashid[1,3], Shumin Liang[1,2], Zulfiqar Ali Sahito[1,2], Ju Yang[4], Zhou Li[4], Zuo Hu[4]*, Zhechao Pan[1,2]*, Qijun Sui[1,2]*

1 Industrial Crops Research Institute, Yunnan Academy of Agricultural Sciences, Kunming, China, 2 Yunnan Technology Innovation Center of Potato (Under Preparation), Kunming, China, 3 Department of Bioinformatics and Biotechnology, Government College University Faisalabad, Faisalabad, Pakistan, 4 Zhaotong Academy of Agricultural Sciences, Zhaotong, China

* zhechaopan@163.com (ZP); huzuo08@163.com (ZH); suiqj@sina.com (QS)

## Abstract

Sustainable agriculture has made significant contributions to both food security and global economic growth. Development of new cultivars and identifying and utilizing the yield potential of existing gemplasms are two key options for sustainable crop production. In this study, field trials and a new plant growth stimulator model developed by the International Potato Center called "Solanum" was used to assess the yield potential and 'yield gaps' of **three potato cultivars in spring, autumn, and early spring seasons in southwestern China**. The results showed that the average potential yield of potato crops in spring, early spring, and autumn seasons was 125.6 t/ha, 56.40 t/ha, and 45.30 t/ha with a gap from the potential yield of 107.30 t/ha, 36.70 t/ha, and 32.10t/ha, respectively. Further analysis revealed that the late blight disease was the main cause of large yield gap in spring season, whereas inadequate rain fall was the the major factor impacting the actual yield of potato crops in autumn and early spring seasons. Therefore, we report for the first time, that the spring potato in Yunnan Province, southwestern China has the highest potential yield in the world, and that extending growing season coupled with managing late blight can increase actual yield by 115%. The high yield potential of spring potato could be very beneficial to the local economy and add an enormous pelf and prosperity to the region.

## Introduction

Potato (*Solanum tuberosum* L.) is the most important non-cereal staple in Asia and South America and serves as a major source of calories and other nutrients for human consumption [1] With the expanding population and resident income, the supply and demand for potatoes are continuously rising [2]. Therefore, the main objective of all crop improvement programs is to increase yield in order to feed the constantly

**Data availability statement:** All relevant data are within the manuscript and its Supporting Information files.

**Funding:** The study was supported by National Key R&D Program Project (2022YFD1601800), National Natural Science Foundation of China (32260505), International Science and Technology Envoy Program (202403AK140090), Yunnan Joint Agricultural Program (202301BD070001-217), Yunnan Talent Project (202305AC160032), Technological Innovation Talent Development Program of Yunnan(202305AD160040), the Agriculture Research System of China (CARS-09-P03), Yunnan Seed laboratory (202205AR070001) and National Talent Demonstration Base (YZJD202001036). The funders had no role in study design, data collection and analysis, decision to publish, or preparation of the manuscript.

**Competing interests:** The authors have declared that no competing interests exist.

increasing world population. Crop production in farmer's field is typically influenced by variety of biophysical factors, aside from the genetic pathways [3].

Yield gap is a common phenomenon which is known as the discrepancy between the actual yield in farmers' fields and the potential yield as predicted by certain spatial and temporal parameters [4]. The potential yield of a cultivar is the yield in its suited habitat with ideal levels of water and nutrients and the efficient management of biotic and abiotic stresses [5]. The International Potato Center (CIP) developed the potato-specific crop model called Solanum in 2012 to simulate the dry mass assimilation and partition for different potato organs following various physiological parameters [6,7]. Using the light interception and utilization (LINTUL) framework, the tuber yield can be precisely predicted [6,8–12]. In addition, this model can predict the changes in potato canopy coverage, fresh tuber weight, and overall biomass depending on the presence or absence of water, nitrogen, and frost settings.

From the global point of view, China has been the largest potato producer in the world. In 2020, 90.321 million metric tons of tubers were harvested from 4.8 million ha land in China, accounting for 27.38% and 30.01% of global potato production and acreage in that year respectively [13]. However, the average yield per land unit production (18.76 ton/ha) in China is still less than the global average (20.94 ton/ha) and considerably less than that achieved in developed countries, which can produce >40 t/ha [14].

Yunnan is one of the largest potato-growing provinces in China, accounting for 66.1% of the planting area in the southwest region (SWC) and contributing to over 35% of the total potato production in China [15]. Yunnan province has a unique complex terrain and topography, associated with its vertical climatic variations and its 94% mountain terrain allows for year round potato production [16,17]. It is the only region in China that has year-round potato production during autumn, early spring, winter and spring seasons. As an important food and economic crop in these areas, the potato planting area was 485,800 hectares, with a total output of about 9.5 million tons in 2014 [18]. The average yield varies from 15.27 t/ha in autumn to 20.72 t/ha in early spring. Despite having adequate amount of rainfall and solar radiation, the average yield in the region is still lower than the global average yield. In order to improve the local potato yield, it is mandatory to determine potato potential yield and recognize the limiting factors in each season. This will help in achieving sustainable economic growth and contribute in financial prosperity for the large mountainous population in Yunnan.

A myriad of sophisticated models are presently deployed globally to forecast the prospective yields of potato cultivars. These cutting-edge tools furnish invaluable insights into the intricate dynamics of potato cultivation, yield potentials, and the intricate interplay with environmental variables. The Agricultural Production System Simulator (APSIM), featuring an integrated potato module, meticulously simulates crop growth trajectories predicated on soil characteristics, meteorological parameters, and agronomic interventions. This model adeptly optimizes resource allocation via astute irrigation scheduling and judicious fertilization regimes, thereby empowering farmers to adopt superior management strategies that augment yields while

curtailing costs. The Decision Support System for Agrotechnology Transfer (DSSAT) amalgamates climatic data, edaphic attributes, and crop management protocols to prognosticate yields and evaluate the repercussions of diverse stressors impinging on potato crops. It serves as a pivotal instrument for farmers, facilitating the formulation of climate change adaptation strategies by gauging the prospective impacts of forthcoming climatic scenarios on potato yields. Concurrently, it proffers expert recommendations on soil fertility stewardship, fostering soil vitality preservation and perpetuating long-term agricultural productivity. Simulation of Turber Cycle under Environmental Conditions (STICS) is exquisitely engineered to mimic the complex growth patterns of the potato crop amidst variegated environmental milieus. This specialized model proves instrumental for farmers, enabling the anticipatory identification of pest and disease pressures, prompting preemptive protective actions. Additionally, it assesses the ramifications of extreme weather phenomena on tuber quality and yield outcomes, enhancing risk mitigation strategies. The Erosion Productivity Impact Calculator (EPIC) delves into the realm of soil erosion processes, intertwined with crop development, inclusive of potatoes. By estimating long-term soil productivity and crop yields, EPIC equips farmers with the knowledge to implement efficacious erosion control mechanisms, safeguard soil integrity and fertility, and embrace sustainable land stewardship practices, thereby ensuring sustained high productivity levels. The Potato Model for Management and Environmental Assessment (POMME) introduces a holistic approach, factoring in tuber maturation stages, water absorption kinetics, and environmental determinants influencing yield. It paves the way for precision irrigation management, striking a balance between maximizing water efficiency and maintaining robust yields. Moreover, POMME's predictive capabilities regarding tuber size distribution facilitate enhanced market planning and grading, optimizing commercial returns. In essence, these avant-garde potato yield models bestow upon farmers a treasure trove of actionable intelligence, enabling them to navigate their cultivation endeavors with heightened acumen. They not only augment productivity and optimize resource utilization but also bolster resilience against adverse elements impinging on potato production. By integrating these state-of-the-art tools into their agricultural frameworks, farmers are poised to reap higher yields, achieve superior resource efficacy, and fortify their capacity to withstand challenges posed by an ever-evolving environment. Compared to other models, the Solanum model has obvious characteristics. It can not only simulate the growth and development process of crops, but also comprehensively consider a variety of factors, such as soil, climate and management measures, to provide a comprehensive solution for agricultural production. The seed model is applicable to different varieties of potatoes and their wild relatives, and adapts to different geo-climatic conditions, and thus has a wide range of applications. The model can dynamically adjust crop growth parameters based on real-time or predicted environmental data, providing more accurate yield predictions. Users can modify the parameters of the model according to specific research objectives or production conditions to make it more realistic.

We hypothesize that (1) Yunnan's Spring potato possesses significant untapped yield potential, primarily constrained by late blight and shortened growth periods, and (2) mitigating these factors can substantially narrow yield gaps. To test this, we employed the Solanum model to conduct a comprehensive assessment of potato yield potential across different seasons in southwestern China. By integrating field trial data with model simulations, we enhanced the model's accuracy and efficacy. This study is the first to quantify the region's largest yield gaps and identify key controlling factors (e.g., disease management, growth period extension). Our findings provide actionable insights for optimizing potato cultivars and natural resource utilization, ultimately improving regional potato production.

## Materials and methods

### Plant materials and experiment sites

The study was conducted at three experimental sites in Yunnan Province, including Yema (103 ° 22′26 ″ E, 26 ° 06′17 ″ N, 2700masl), Songming (103 ° 06 ′E, 25 ° 21′380 ″ N, 1917masl), and Chaotie (103 ° 40′12 ″ E, 25 ° 01′48 ″ N, 1840masl) (S1 Table). These locations correspond to spring, autumn, and early spring cropping seasons. Eight commercially important cultivars and diverse characteristics were assessed for the actual and potential yields of potato crops in Yunnan.

Three well-adapted cultivars Yunshu 505 (early maturing), Yunshu 401 (medium maturing), and Cooperation 88 (Late maturing) were selected to assess yield potential, whereas, Yunshu 401, Yunshu 103 (early maturing), S03-905, Yunshu 503 (medium maturing), and S05-277 (late maturing) breeding lines were used to examine the effect of late blight for gap mitigation. The characteristics of these potato cultivars are described in Table 1.

## Experimental design and technical approach

The whole tubers were sown and regionally standardized fertilizers were used as: 15–18 ton/ha of farmhouse fertilizer, 195 Kg/ha of Urea, 1125 Kg/ha of superphosphate, and 600 Kg/ha of potassium sulfate compound fertilizers. During the growth periods of 2013–2015, disease free and undamaged whole tubers weighing with the minimum weight of 100 g were sown using a randomized complete block design (RCBD) with three replications. In order to explore the potential yields of the three breeding lines, each breeding line was planted in seven plots, each with four rows and five plants in each row to make 21 experimental units per replication (S1 Fig). Whereas, in order to study the impact of late blight, each genotype was planted in six plots with eight rows in each plot. The row spacing and plant to plant spacing were kept constant at 67 cm and 30 cm respectively. The complete growth cycles for the three cultivation seasons were characterized as follows: early spring potatoes (January-May, ~150 days), spring potatoes (March-September, ~180 days), and autumn potatoes (August-December, ~120 days). This distinct seasonal partitioning reflects Yunnan's unique capacity for year-round potato production enabled by its vertical climatic zonation.

## Estimation of Parameters and potential yield

To calculate the maximum potential yield parameters for each genotype, precisely data were recorded through continuous field survey: sowing time, harvest time, planting density (plants/m [2]), 50% emergence date, maximum canopy coverage index, days to maximum canopy cover, date of 50% death above ground, whole growth period in days, and water and fertilizer management. The meteorological data specifically for maximum temperature ($T_{max}$), minimum temperature ($T_{min}$), total solar radiation (SR), rainfall and evaporation were collected by a small weather station (MLS-1306) installed at each experimental site.

A digital camera was used to record the canopy covering. No zoom, no flash, ISO100, and maximum resolution were all set the on the camera before taking the images. Three pictures were taken for each genotype from the experimental unit, a total 63 photos of each genotype each time. The camera lens was held horizontal to the ground at a height of 60 cm while capturing pictures. These images were taken every 10–15 days following the emergence of the seedlings. MATLAB software was used to analyze the resulting images and extract canopy coverage information. Then these canopy coverage data were fit to canonical quadratic equation to calculate the number of days required to achieve 1% canopy coverage.

Once the tube planting started, the yield was measured every 10–15 days. Meanwhile, plots from each replication were randomly selected for each genotype for dry weight estimation. Moreover, samples from the 12 protected plants were

**Table 1. The list of locally adapted varieties used in the study and their characteristics.**

| Variety | Trait | Maturity | Breeding Institute |
|---|---|---|---|
| Yunshu 505 | Oval, yellow skin and white flesh | E | ICRI, YAAS |
| Yunshu 401 | Long-shaped, white skin and white flesh | M | ICRI, YAAS |
| Cooperation 88 | Oval, Red skin and white flesh | L | Cooperation breeding |
| Yunshu 103 | Oval, White skin and White flesh | E | ICRI, YAAS |
| Yunshu 503 | Round, White skin and white flesh | M | ICRI, YAAS |
| S03-905 | Oval, White skin and White flesh | M | ICRI, YAAS |
| S05-277 | Long-shaped, Red skin and red ring flesh | L | ICRI, YAAS |

taken, and their roots, stems, leaves and tubers were weighed separately to estimate fresh and dry weight. The plant harvest index (HI) is estimated with the following equation:

$$Harvest\ Index = \frac{Dry\ mass\ contents\ of\ tuber}{Whole\ plant\ dry\ weight}$$

(1)

The Intercepted Photosynthetically Active Radiation (IPAR) is one of the parameters used to evaluate the potential yield. Hence, the daily total solar radiation data were collected from meteorological stations, and Photosynthetically Active Radiation (PAR) were calculated as the 50% of the total photosynthetic radiation for total canopy coverage (cc) at each day (d) until harvest (N). Finally, the IPAR was estimated with following formula:

$$IPAR = \sum_{d=1}^{N} PAR \times (0.5) \times cc$$

(2)

Further radiation use efficiency (RUE) was calculated by the linear relationship between the IPAR and total potato biomass.

Another common method of estimating the potential yield for each genotype is to use the potato block expansion period (PBEP), which is the period of time until 50% of the plants produce at least one potato block tuber exceeding in the diameter of 1.0 cm.

The Solanum-tool estimates crop parameters using allometric and heuristic methods based on the relationship between the aerial and tuber partitioning crop growth functions. The canopy cover evaluation was performed by adopting Beta function as shown in Eq. (3) [19], and the Gompertz function as shown in Eq. (4) was used to assess tuber partitioning over time [20]. The minimal partition function was considered to be the start of tuber initiation. The parameters were calculated using the following two equations:

Beta function:

$$W = W_{max} \left(1 + \frac{te - t}{te - tm}\right) \left(\frac{t}{te}\right) \frac{te}{(te - tm)} \text{ with } 0 \leq tm < te$$

(3)

$W_{max}$ is the maximum coverage, $t_m$ is the thermal time at the maximum canopy cover growth rate, and $t_e$ is thermal time at maximum canopy cover,

Gompertz function:

$$Y = A \times e^{(-e^{(-\frac{t-Tu}{b})})}$$

(4)

where "A" is the maximum harvest index, Tu is the thermal time at maximum tuber partition rate, and b is the thermal time just before the tuber initiation process.

The thermal time value (TT) a parameter defining the sunlight absorption foreach day of the entire growth period and calculated by the emergence days, maximum canopy coverage maintenance days, and maturity days. The beta algorithm can be used to determine the linear relation between the thermal time and canopy cover values, using Gompertz function to combine the dry matter content and the potential yield can be estimated.

## Estimation of the effect of late blight on potential yield

To evaluate the effect of late blight on potential yield, the area under disease progress curve (AUDPC) was calculated from the estimated percentages of infected leaf area at different times during the epidemics. The AUDPC can be calculated using the following midpoint formula:

$$AUDPC = \sum_{i=1}^{n-1} \left(\frac{y_i + y_{i+1}}{2}\right) (t_{i+1} - t_i)$$

(5)

where "t" is the time of $i^{th}$ reading, "y" is the percentage of affected foliage at $i^{th}$ reading and "n" is the number of readings. The variable "t" represents the Julian days, days after planting or days after emergence.

For the prevention and control of late blight, fungicide was applied to the foliage (Yi Bao, Yi Kuaijing and Yin Fali manufactured by Du Pont, USA and Bayer, Germany) on 1st, 15th and 29th of August 2013, and 30th June, 22nd July, 5th, 14th and 27th August, and 17th September of 2014.

## Data processing

The SAS program was used to estimate the crop model parameters and performed analysis of variance. The MATLAB software was used for image processing to acquire canopy converge. The potential yield was estimated by Solanum V3.05. The Origin Pro 7.0 was used to obtain the logistic growth curves for tuberization rate, and canopy coverage fitting.

## Results

### Analysis of meteorological data and climatic conditions during the growing period

The meteorological data were continuously recorded at each experimental unit, including temperature, rainfall, solar radiation, and daily sunshine hours. It was observed that the spring potato has the longest growing period which lasts from March to September (up to 200 days) in Southwestern China, followed by the early spring potato lasting from January to May (130 days), and the autumn potato lasting from August to December (120 days). The maximum temperature in the three seasons ranges from 19.4°C to 21.8°C, 13.7°C to 24.1°C, and 14.5°C to 27.6°C respectively (Table 2). The temperature difference between the morning and evening temperatures in these three seasons was 6.9°C to 11.1°C, 7.8°C to 10.62°C, and 12.4°C to 15.5°C respectively. The similar trend was observed for solar radiation (13.3 to 20.5 MJ· m⁻², 15.74 to 23.7 MJ· m⁻², and 8.9 to 13.1MJ m⁻²), and sunshine hours (11.8 to 13.6 h, 10.8 to 13.4 h, and 10.5 to 12.2 h) during the three growth seasons respectively. The highest rainfall (1149.2 mm) was observed in spring, but less in autumn (362.4 mm), and early spring (190.1 mm). The monthly rainfall in spring was unevenly distributed throughout the growth period, which ranges from 14.2 mm in early growth season (March) to 354.4 mm in July. Early spring potatoes experienced a dry seedling stage due to the scant rainfall (5.2 mm) in February, while at later growth stages during November-December observed minor precipitation. As shown in Table 2, the pattern of rainfall in autumn was reversed, with higher rainfall appearing in early growth (180.4 mm in August), and less in later phases (15.8 mm in November, 35.4 mm in December).

### Emergence and canopy coverage

According to the field data, spring potatoes have the highest canopy coverage (99%) and required the longest maintenance period (143 days), followed by autumn potato (97%, 81 days), and early spring potato (81%, 104 days)

**Table 2. Meteorological data during different growing periods of potato.**

| | Spring | | | | | | | Autumn | | | | | Early Spring | | | | |
|---|---|---|---|---|---|---|---|---|---|---|---|---|---|---|---|---|---|
| | Mar | Apr | May | Jun | Jul | Aug | Sep | Aug | Sep | Oct | Nov | Dec | Jan | Feb | Mar | Apr | May |
| $T_{max}$ (°C) | 19.5 | 21.8 | 21.3 | 19.6 | 20.3 | 19.4 | 20.1 | 23.9 | 24.1 | 20.9 | 19.22 | 13.7 | 14.5 | 18.7 | 24.6 | 23 | 27.6 |
| $T_{min}$ (°C) | 10.5 | 11.8 | 10.2 | 12.6 | 12.3 | 12.1 | 11.3 | 16.1 | 16.3 | 11.4 | 8.6 | 5.1 | 1.2 | 3.2 | 8.5 | 10.6 | 13.9 |
| $T_{dif}$ | 9 | 10 | 11.1 | 6.9 | 8.0 | 7.3 | 8.9 | 7.8 | 7.8 | 9.5 | 10.62 | 8.6 | 13.3 | 15.5 | 16.1 | 12.4 | 13.7 |
| Rainfall | 14.2 | 21.4 | 128 | 236.2 | 354.4 | 150.2 | 244.8 | 180.4 | 41.4 | 89.4 | 15.8 | 35.4 | 72.5 | 5.2 | 26 | 70.8 | 15.6 |
| SR | 19.8 | 20.5 | 19.8 | 13.3 | 14.3 | 15.2 | 17.5 | 12.7 | 11.1 | 12.9 | 13.1 | 8.9 | 23.7 | 15.74 | 19.3 | 16.9 | 20 |
| SnH | 11.8 | 12.6 | 13.3 | 13.6 | 13.4 | 12.9 | 12.1 | 12.2 | 12.1 | 11.4 | 10.8 | 10.5 | 13.4 | 12.9 | 12.1 | 11.4 | 10.8 |

Note: $T_{max}$ is the maximum temperature, $T_{min}$ is the minimum temperature, $T_{dif}$ indicates the temperature difference between morning and evening in°C, Rainfall in mm, SR is solar radiation measured in MJ·m⁻²·d⁻¹, and SnH is sunshine hours.

respectively. Spring has the longest growing season; potato tuber emergence was relatively delayed (88–102 days after sowing). With the early emergence (16–25 days after sowing) in autumn the tuber initiation was also early (39–49 days after sowing). A very late emergence (65 days after sowing) with relatively early tuber initiation in early spring potato was noted (Table 3).

## Model parameters

According to the model estimates, the maximum canopy cover in spring season was typically 0.999, which was consistent with that observed in field survey shown in (Tables 3 and 4). This value was greater than that in autumn (0.92), and early spring potato (0.80). The similar trend was observed for the thermal time just before the tuber initiation (b), the thermal time at tuber partition rate (Tu), and the thermal time at end of the growth period (te). These variables work together to produce the same radiation use efficiency (RUE) pattern (Spring 4.62 > autumn 3.13 > early spring 2.71). The average thermal time at maximum canopy cover (tm) was higher in autumn season (449°C) than in spring (256°C), and early spring (192°C) seasons. While the average tuber partition value (A) showed a different trend, peaking at (0.97) in the early spring and falling to (0.91) in autumn as shown in (Table 4).

**Table 3. Field survey data for different crop seasons of potato.**

| Seasons | Genotypes | Days to Emergence | Maximum canopy cover index | Maximum harvest index | Days to reach 1% canopy cover | Days to maximum canopy cover | Days to physio-logical maturity | Days to Tuber initiation |
|---------|-----------|-------------------|----------------------------|-----------------------|-------------------------------|------------------------------|---------------------------------|--------------------------|
| Spring | YunShu 401 | 47 | 0.98 | 0.82 | 52 | 143 | 170 | 99 |
| | Yunshu 505 | 44 | 0.99 | 0.97 | 51 | 143 | 164 | 88 |
| | Cooperate88 | 42 | 0.98 | 0.85 | 44 | 107 | 184 | 102 |
| Autumn | Yunshu 401 | 21 | 0.78 | 0.98 | 21 | 68 | 107 | 39 |
| | Yunshu 505 | 25 | 0.96 | 0.87 | 25 | 81 | 120 | 49 |
| | Cooperate88 | 16 | 0.97 | 0.87 | 16 | 81 | 116 | 45 |
| Early spring | Yunshu 401 | 34 | 0.77 | 0.956 | 40 | 104 | 123 | 73 |
| | Yunshu 505 | 32 | 0.81 | 0.973 | 37 | 104 | 121 | 71 |
| | Cooperate88 | 35 | 0.81 | 0.976 | 37 | 104 | 122 | 72 |

**Table 4. Comparative assessment of model parameters estimated by parameters estimator.**

| Season | Genotype | Wmax | tm | te | A | tu | b | RUE | Y |
|--------|----------|------|-----|------|--------|-------|-------|------|---|
| Spring | Yunshu 401 | 1 | 243 | 1145 | 0.88 | 988 | 426 | 5.5 | 138 ± 9.6A |
| | Yunshu 505 | 0.99 | 99 | 1107 | 1 | 1044 | 473 | 4.4 | 147 ± 10.7A |
| | Cooperate88 | 1 | 427 | 1304 | 0.89 | 1258 | 394 | 3.98 | 91.9 ± 10.3B |
| Autumn | Yunshu 401 | 0.77 | 482 | 1040 | 0.96 | 669 | 290 | 2.9 | 35.23 ± 2.3C |
| | YS505 | 1 | 341 | 1014 | 0.91 | 721 | 250 | 2.8 | 46.67 ± 3.1B |
| | Cooperate88 | 1 | 524 | 1028 | 0.87 | 823 | 190 | 3.7 | 53.95 ± 3.6A |
| Early spring | Yunshu 401 | 0.7742 | 200.1 | 956.4 | 0.9249 | 544 | 150.1 | 2.8 | 53.03 ± 4.2B |
| | Yunshu 505 | 0.83 | 112.7 | 898.5 | 0.99 | 545.8 | 186.5 | 2.63 | 62.78 ± 4.46A |
| | Cooperate88 | 0.8 | 262 | 946 | 1 | 532 | 386 | 2.7 | 53.48 ± 4.09B |

Note: Wmax is the maximum canopy coverage; Tm is the thermal time at maximum canopy cover rate; te is the thermal time at end of growth period; A is the maximum value of tuber partition; Tu is the thermal time at tuber partition rate; b is the thermal time just before the tuber initiation; Y is the Yield Simulated on potential conditions (t/ha).

## Magnitude of Potential yield and yield gap at southwestern China

The estimated potential yield from the Solanum model is compared to the actual yield observed from farmers' field [18]. Disregarding the genotypes and sites, the spring potato has the maximum potential yield (125.6 t/ha) followed by early spring potato (56.4 t/ha), and autumn (45.3 t/ha) potato. Although the actual yields in spring (19.6 t/ha) and early spring (20.72 t/ha) are not significantly different, there is a huge difference in potential yields between these two seasons. The largest yield gap is 106 t/ha in spring potato followed by the early spring potato (35.68 t/ha). As with the potential yield, the actual yield in autumn season is also lower, resulting the lowest yield gap (30.03 t/ha). Breeding lines differ significantly in yield potential among these seasons. The genotypic performance of breeding lines at three experimental locations might be differentiated in addition to substantial environmental variables. The Yunshu-505 was the best performing cultivar with the maximum potential yields in spring (147 t/ha), and early spring (62.78 t/ha). The Cooperation 88 showed best performance with maximum potential yield in autumn season (53.95 t/ha) and the minimum potential yield in other seasons (Table 4, Fig 1).

## Effect of late blight control on narrowing yield gap in spring season

The results revealed that spring, autumn and early spring potatoes in Yunnan have high yield potential and huge gaps between the actual and potential yields were observed. It is well known that the areas under study have cold environment with concentrated rainfall which are ideal conditions for late blight development and makes the late blight main factor, contributing the large yield gaps, particularly in spring potatoes. To estimate its output reduction effect, the late blight in five varieties was monitored in 'controlled' and 'not controlled' (after artificial inoculation) treatment plots. The sampling and evaluation for disease severity were performed seven times during the growth period. The impact of late blight on potato yield was evaluated using curve fitting of these results.

A significantly negative association was observed between the disease index of late blight and tuber yield by regression analysis ($R^2 = 0.69$) (Fig 2). Moreover, the area under disease progress curve (AUDPC) dramatically decreased after late blight was controlled and overall yield increased (Fig 3). It can be seen that the fitted results are consistent with the logistic growth curve (Fig 3). The results of the tuber initiation in these crop seasons showed that the late blight control could significantly increase the yield by 42.73% (S03-905) to 115.11% (Yunshu 401). As shown in Fig 4, the prevention and control of late blight delayed the maximum rate of potato initiation. It was observed that the fastest rate of tuber initiation

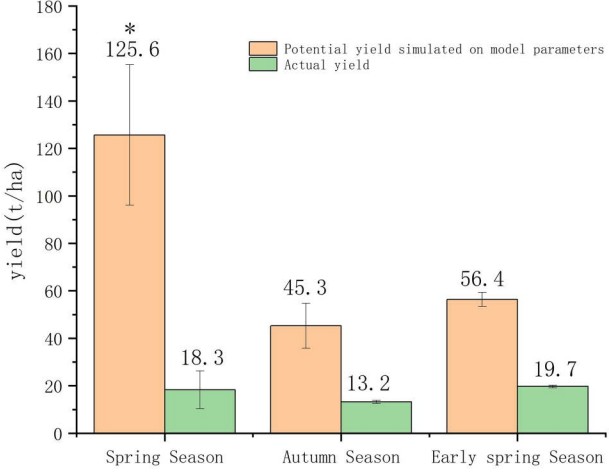

**Fig 1. Comparison of actual and potential yields and yield gap in different crop seasons.**

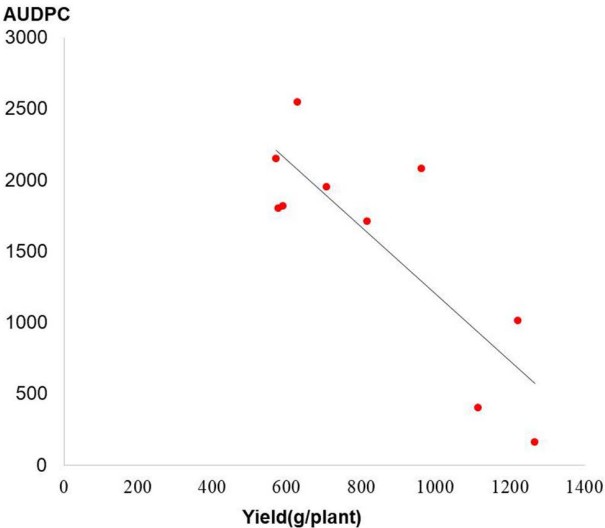

**Fig 2. Regression analysis between yield and disease index among different potato varieties, where R [2]=0.6904, and y=-2351x+3553.**

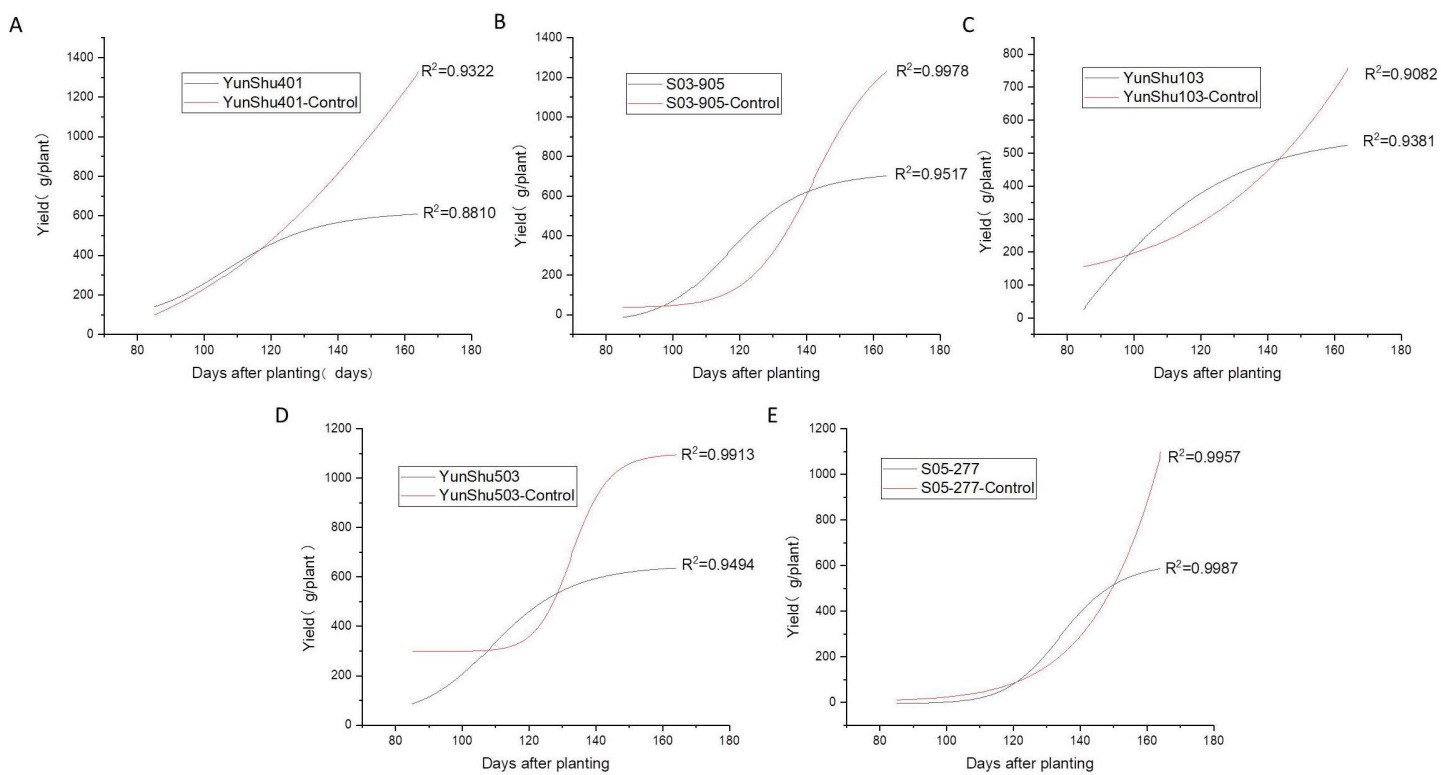

**Fig 3. Yield fitting curve for different potato varieties with and without fungicide application to prevent and control the late blight disease.**

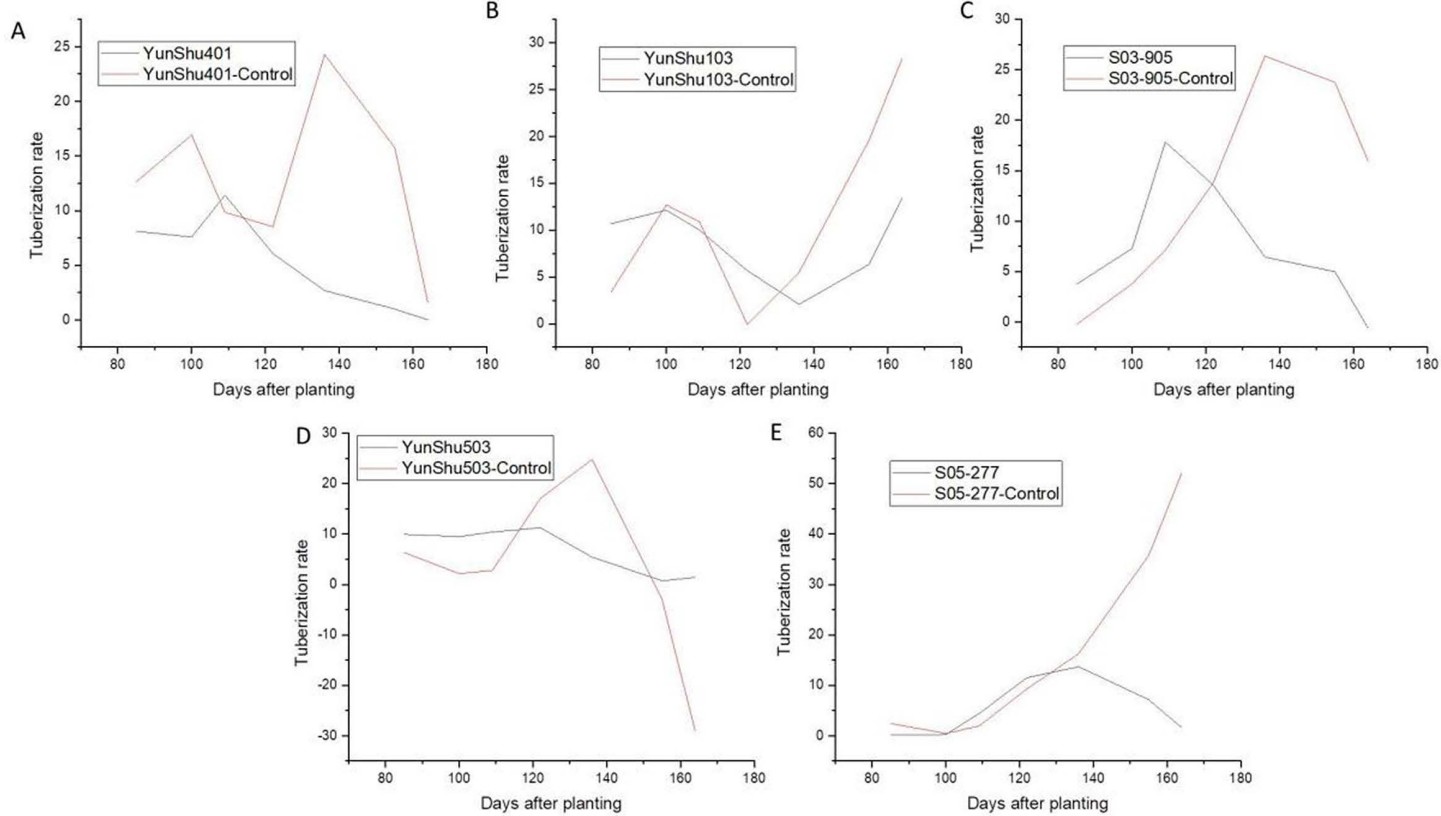

**Fig 4. Evaluation of tuber initiation period of different potato varieties with and without fungicide application to prevent and control the late blight disease.**

appearedin medium maturing genotypes Yunshu 401, S03-905, and Yunshu 503, and due to prevention and control, the late blight delayed for 27 days and shifted from 2nd August to 29th August. In addition, in the case of early maturing Yunshu 103 or late maturing S05-277 genotypes, through controlling the late blight, the total yield wasimproved without any delay in the rate of maximum tuber initiation. These findings supported the hypothesis that disease occurrence led to the termination of potato growth cycle and reduced potato yield. Whereas, disease prevention and control, extended the potato growing period and the yield was significantly increased.

## Discussion

Food security fundamentally entails reliable access to sufficient, affordable, and nutritious food supplies [21]. This study demonstrates that through analyzing yield potential gaps [15] coupled with environmental modulation [22], China's primary potato-producing regions could achieve a potential yield of 160 t/ha [23]. However, climate change-induced shortening of growth periods and inefficient water-fertilizer management have significantly constrained yield potential realization.

Our findings reveal that cultivar Yunshu 505 cultivated in Yunnan's spring season demonstrates a potential yield of 147 t/ha, yet actual yields remain below 30 t/ha, representing the largest documented yield gap globally (Fig 1). This aligns with FAO agro-ecological zone estimates for Southwest China (134.2 ± 9.0 t/ha) [24]. Comparative data shows Sub-Saharan Africa with merely 65 t/ha potential yield and 5 t/ha yield gap, while Pakistan reports 40–50 t/ha (spring) and 34–47 t/ha (autumn) yields, highlighting Yunnan's distinctive advantages for spring potato production.

Optimal growing conditions are evidenced by abundant light-temperature resources (mean daily radiation 15 MJ·m$^{-2}$·d$^{-1}$), 1100 mm precipitation, and favorable diurnal temperature variations (Table 2). Nevertheless, excessive humidity during July-August precipitates late blight epidemics, causing premature growth termination and reduced photoassimilate accumulation. Fungicide applications effectively extended the growth period by 27 days, yielding 115% production increases (Figs 3 and 4).

Autumn cultivation faces compounded limitations: 362.4 mm seasonal precipitation (50% occurring post-tuberization) coupled with modest 11.74 MJ·m$^{-2}$·d$^{-1}$ mean radiation. Early spring crops confront acute water constraints, with merely 5.2 mm precipitation during emergence (190.1 mm total cycle) despite high radiation inputs (19.13 MJ·m$^{-2}$·d$^{-1}$).

Consequently, Yunnan's spring potato production strategy should prioritize: (1) developing late blight-resistant cultivars combined with growth period extension techniques, achieving verified yields of 97.2 t/ha (Zhaotong field trials, 2021); and (2) adopting mulching water conservation techniques for early spring/autumn cultivation to mitigate hydrological constraints.

## Conclusion

The potential yields of potatoes in Yunnan predicted by combining the Solanum model and its actual yields are consistent with those estimated by using the light-temperature potential model of the FAO agro-ecological region method, which proves the accuracy, and the feasibility of the present method. Yunnan has a favorable production environment, and the availability of natural resources like as rainfall, mild temperatures, and high-altitude solar radiation provide the great potential for further growth benefits for the spring potato crops. The selection of healthy seed potatoes, plantation of disease-resistant varieties and disease control to ensure nutrient supply in the late stage of crop growth, and the extended growth period up to the end-September can make full use of natural resources, which is an effective measure to increase the yield of spring potatoes and to narrow the yield gap from the potential value. On November 15, 2021, the field production test conducted by our unit in Zhaotong City, the main potato producing area of spring potato, Yunnan Province, showed that the field harvested potatoes were 672.18 m$^2$, and the actual yield was 97.20 t/ha. This study demonstrated that the potential yield of spring potato in southwest China very high and improvement in cultivation practices, even higher yield can be achieved. The bottleneck for early spring and autumn potato production is the small amount of rainfall. Therefore, using appropriate water-saving mulching techniques is the key to raising their yields.

## Supporting information

**S1 Fig. Planting pattern for each potato genotype randomized in experimental plots, each block represents an experimental plot, A indicates the genotype, and the number besides A is the replication, while B is the harvest frequency in each block.**
(JPG)

**S1 Table. Soil Information of the three experimental sites Yema, Songming, and Chaotie.**
(DOCX)

## Acknowledgments

The authors are thankful to the International Potato Center (CIP) and China-CIP Center for Asia Pacific (CCCAP) for provision of the Solanum model.

## Statement and declaration

The authors declare no competing interest.

## Author contributions

**Conceptualization:** Zhechao Pan, Qijun Sui.

**Data curation:** Muhammad Abdul Rehman Rashid.

**Funding acquisition:** Qijun Sui.

**Investigation:** Ying Wang, Shumin Liang, Zhechao Pan, Ju Yang, Zhou Li, Zuo Hu.

**Methodology:** Ying Wang.

**Software:** Ying Wang, Shumin Liang.

**Supervision:** Zhechao Pan, Qijun Sui.

**Writing – original draft:** Ying Wang, Muhammad Abdul Rehman Rashid.

**Writing – review & editing:** Zhechao Pan, Qijun Sui, Zulfiqar Ali Sahito.

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
