## [Decision Letter · Decision Letter 0]

25 Sep 2024

PONE-D-24-36013Improving the prediction of Potato yield gaps: Solanum-model parameterization and evaluation in southwestern ChinaPLOS ONE

Dear Dr. Wang,

Thank you for submitting your manuscript to PLOS ONE. After careful consideration, we feel that it has merit but does not fully meet PLOS ONE’s publication criteria as it currently stands. Therefore, we invite you to submit a revised version of the manuscript that addresses the points raised during the review process.

We look forward to receiving your revised manuscript.

Kind regards,

Tzen-Yuh Chiang

Academic Editor

PLOS ONE

**Journal Requirements:**

Yunnan Academy of Agricultural Sciences Scientific Research Pre-research Project (2023KYZX-05) and Yunnan Joint Agricultural Program (202301BD070001-217)

6. Please ensure that you refer to Figure 4 in your text as, if accepted, production will need this reference to link the reader to the figure.

7. Please include your tables as part of your main manuscript and remove the individual files. Please note that supplementary tables (should remain/ be uploaded) as separate "supporting information" files.

Reviewers' comments:

Reviewer's Responses to Questions

**Comments to the Author**

1. Is the manuscript technically sound, and do the data support the conclusions?

Reviewer #1: No

2. Has the statistical analysis been performed appropriately and rigorously? 

Reviewer #1: No

3. Have the authors made all data underlying the findings in their manuscript fully available?

Reviewer #1: No

4. Is the manuscript presented in an intelligible fashion and written in standard English?

Reviewer #1: No

5. Review Comments to the Author

**Reviewer #1: ** Introduction should be written again. It is very short and most of the information is related to geography and yield. It is better if author include more review about the different models being used in the world and how they are beneficial for farmers. Hypothesis is missing. Rephrase the objective of the studies.

Materials and methods:

No weather and soils information are given about the studied locations.

Ten years old research During the growth periods of 2013-2015.

Line 94-96: The total growth period 94 from planting to harvesting for the early spring, spring and autumn seasons, spanned from early January to the end of 95 May, mid-March to end-of-September, and mid-August to mid-December for early-spring, spring, and autumn seasons, 96 respectively.

This sentence should be written again.

No tables are provided in the PDF file.

The paper is not written according the PLOS one standard.

Authors is not clear about the results.

Discussion is very weak.

Not recommended.

6. PLOS authors have the option to publish the peer review history of their article (what does this mean? ). If published, this will include your full peer review and any attached files.

**Do you want your identity to be public for this peer review?** For information about this choice, including consent withdrawal, please see our Privacy Policy .

Reviewer #1: No

---

## [Author Response · Author response to Decision Letter 1]

21 May 2025

Comment: 1. Introduction should be written again. It is very short and most of the information is related to geography and yield. It is better if author include more review about the different models being used in the world and how they are beneficial for farmers. Hypothesis is missing. Rephrase the objective of the studies.

Response: We sincerely thank the reviewer for the constructive feedback on the Introduction section. As suggested, we have thoroughly revised the Introduction to improve its depth and clarity. Specifically, we have: added a comprehensive review of globally used crop simulation models (e.g., APSIM, DSSAT, STICS, EPIC, and POMME), highlighting their relevance and benefits to farmers in terms of yield optimization, resource management, and climate resilience. Clearly stated the hypothesis of the study, focusing on the role of biotic and abiotic factors in contributing to potato yield gaps in Yunnan. Rephrased and clarified the objective of the study to better reflect the goals of quantifying potential yield, identifying limiting factors, and evaluating the Solanum model's applicability for enhancing yield outcomes. Now, we believe these revisions have significantly strengthened the scientific context and relevance of the study.

Comment: 2. Materials and methods:

No weather and soils information are given about the studied locations.

Response: Thank you for pointing out that weather information is misssing. In response to your valuable suggestion, we have now included detailed meteorological data—covering temperature, rainfall, solar radiation, and sunshine hours—for each cropping season and experimental location in Table 2 of the revised manuscript. Additionally, we have provided the soil characteristics of the three experimental sites (Yema, Songming, and Chaotie) in Supplementary Table S1. These additions enhance the environmental context of the study and support the interpretation of the model outputs and yield results.

Comment: 3: Ten years old research During the growth periods of 2013-2015.

Response: We acknowledge the reviewer’s observation regarding the period of the field experiments (2013–2015). While the data were collected nearly a decade ago, they remain highly relevant for several reasons:

1.Baseline Model Calibration: The purpose of the study is to evaluate the Solanum model's parameterization under well-documented field conditions. The historical dataset provides strong and detailed agronomic, phenological, and environmental information essential for validating and improving the model's accuracy.

2.Climatic Stability of the Region: Yunnan Province exhibits relatively stable climatic patterns over the years, particularly in terms of temperature and altitude-dependent cropping systems. Thus, the findings remain applicable to current cultivation practices.

3.Limited Shifts in Cultivar Use: The cultivars studied are still widely grown or serve as the genetic basis for more recent breeding lines, making the analysis and conclusions still pertinent to ongoing production systems.

4.Model Utility Over Time: Since crop simulation models like Solanum are meant to provide generalizable insights across years and conditions, their calibration using historical datasets enhances future projections and current applicability.We agree that future studies should aim to include more recent datasets and are planning to validate the model under updated climate scenarios and recent field data in ongoing research.

Comment: 4: Line 94-96: The total growth period 94 from planting to harvesting for the early spring, spring and autumn seasons, spanned from early January to the end of 95 May, mid-March to end-of-September, and mid-August to mid-December for early-spring, spring, and autumn seasons, 96 respectively. This sentence should be written again.

Response: Thank you for the suggestion. We have revised the sentence and marked in blue color. “The complete growth cycles for the three cultivation seasons were characterized as follows: early spring potatoes (January-May, ~150 days), spring potatoes (March-September, ~180 days), and autumn potatoes (August-December, ~120 days). This distinct seasonal partitioning reflects Yunnan's unique capacity for year-round potato production enabled by its vertical climatic zonation.”.

Comment: 5: No tables are provided in the PDF file.

Response: We have included tables in the PDF

Comment: 6: The paper is not written according the PLOS one standard.

Response: Thank you for suggestion. We have thoroughly revised the manuscript and written according to the PLOS one standard.

Comment: 6: Authors is not clear about the results.

Response: Thank you for your valuable feedback. We have carefully reviewed the manuscript and thoroughly revised it, and ensured that the presentation of our results is clear and precise. We believe that the revised version addresses the concerns raised, and we have made efforts to clarify our findings and avoid any uncertainty. We hope that the revised manuscript reveals our clarity on the results and provides a more clear narrative. If further clarification is needed, we are happy to make additional revisions." Thank you for the comments.

Comment: 6: Discussion is very weak.

Response: Thank you for your constructive feedback regarding the discussion section. We appreciate your insights, we have thoroughly revised the discussion to better interpret the results and relate them to the existing literature. We have also provided additional context and analysis to strengthen our conclusions and clarify the implications of our findings. We believe that these revisions enhance the overall quality of the discussion, and we hope the updated section now meets the expectations. Should you have further suggestions or points for improvement, we are happy to address them."

---

## [Decision Letter · Decision Letter 1]

7 Jul 2025

Improving the prediction of Potato yield gaps: Solanum-model parameterization and evaluation in southwestern China

PONE-D-24-36013R1

Dear Dr. Wang,

We’re pleased to inform you that your manuscript has been judged scientifically suitable for publication and will be formally accepted for publication once it meets all outstanding technical requirements.

Kind regards,

Tzen-Yuh Chiang

Academic Editor

PLOS ONE

Additional Editor Comments (optional):

Reviewers' comments:

Reviewer's Responses to Questions

**Comments to the Author**

1. If the authors have adequately addressed your comments raised in a previous round of review and you feel that this manuscript is now acceptable for publication, you may indicate that here to bypass the “Comments to the Author” section, enter your conflict of interest statement in the “Confidential to Editor” section, and submit your "Accept" recommendation.

Reviewer #1: All comments have been addressed

2. Is the manuscript technically sound, and do the data support the conclusions?

Reviewer #1: Yes

3. Has the statistical analysis been performed appropriately and rigorously? 

Reviewer #1: Yes

4. Have the authors made all data underlying the findings in their manuscript fully available?

Reviewer #1: Yes

5. Is the manuscript presented in an intelligible fashion and written in standard English?

Reviewer #1: Yes

6. Review Comments to the Author

Reviewer #1: The revised manuscript is in good shape and author addressed all the required concerned in the revised version.

7. PLOS authors have the option to publish the peer review history of their article (what does this mean? ). If published, this will include your full peer review and any attached files.

**Do you want your identity to be public for this peer review?** For information about this choice, including consent withdrawal, please see our Privacy Policy .

Reviewer #1: **Yes: ** Sajjad Hussain

---

## [Editor Report · Acceptance letter]

PONE-D-24-36013R1

PLOS ONE

Dear Dr. Wang,

I'm pleased to inform you that your manuscript has been deemed suitable for publication in PLOS ONE. Congratulations! Your manuscript is now being handed over to our production team.

You will receive an invoice from PLOS for your publication fee after your manuscript has reached the completed accept phase. If you receive an email requesting payment before acceptance or for any other service, this may be a phishing scheme. Learn how to identify phishing emails and protect your accounts at >https://explore.plos.org/phishing.

Kind regards,

on behalf of

Dr. Tzen-Yuh Chiang

Academic Editor

PLOS ONE